# The Value of Heart Rhythm Complexity in Identifying High-Risk Pulmonary Hypertension Patients

**DOI:** 10.3390/e23060753

**Published:** 2021-06-15

**Authors:** Shu-Yu Tang, Hsi-Pin Ma, Chi-Sheng Hung, Ping-Hung Kuo, Chen Lin, Men-Tzung Lo, Hsao-Hsun Hsu, Yu-Wei Chiu, Cho-Kai Wu, Cheng-Hsuan Tsai, Yen-Tin Lin, Chung-Kang Peng, Yen-Hung Lin

**Affiliations:** 1Department of Internal Medicine, National Taiwan University Hospital and National Taiwan University College of Medicine, Taipei 100, Taiwan; y03833@ms1.ylh.gov.tw (S.-Y.T.); 009578@ntuh.gov.tw (C.-S.H.); kph712@ntuh.gov.tw (P.-H.K.); chokaiwu@ntuh.gov.tw (C.-K.W.); 2Department of Internal Medicine, National Taiwan University Hospital Yun-Lin Branch, Yun-Lin 640, Taiwan; 3Department of Electrical Engineering, National Tsing Hua University, Hsinchu 300044, Taiwan; hp@ee.nthu.edu.tw; 4Department of Biomedical Sciences and Engineering, National Central University, Taoyuan City 330, Taiwan; clin@ncu.edu.tw (C.L.); mzlo@ncu.edu.tw (M.-T.L.); 5Department of Surgery, National Taiwan University Hospital and National Taiwan University College of Medicine, Taipei 100, Taiwan; 008295@ntuh.gov.tw; 6Department of Computer Science and Engineering, Yuan Ze University, Taoyuan City 330, Taiwan; dtmed005@saturn.yzu.edu.tw; 7Cardiology Division of Cardiovascular Medical Center, Far Eastern Memorial Hospital, New Taipei City 220, Taiwan; 8Department of Internal Medicine, National Taiwan University Hospital Jin-Shan Branch, New Taipei City 220, Taiwan; 9Department of Internal Medicine, Taoyuan General Hospital, Taoyuan City 330, Taiwan; 10Division of Interdisciplinary Medicine and Biotechnology, Beth Israel Deaconess Medical Center/Harvard Medical School, Boston, MA 02215, USA; ckpeng@comcast.net

**Keywords:** pulmonary hypertension, heart rate variability, non-linear analysis, detrended fluctuation analysis, multiscale entropy

## Abstract

Pulmonary hypertension (PH) is a fatal disease—even with state-of-the-art medical treatment. Non-invasive clinical tools for risk stratification are still lacking. The aim of this study was to investigate the clinical utility of heart rhythm complexity in risk stratification for PH patients. We prospectively enrolled 54 PH patients, including 20 high-risk patients (group A; defined as WHO functional class IV or class III with severely compromised hemodynamics), and 34 low-risk patients (group B). Both linear and non-linear heart rate variability (HRV) variables, including detrended fluctuation analysis (DFA) and multiscale entropy (MSE), were analyzed. In linear and non-linear HRV analysis, low frequency and high frequency ratio, DFAα1, MSE slope 5, scale 5, and area 6–20 were significantly lower in group A. Among all HRV variables, MSE scale 5 (AUC: 0.758) had the best predictive power to discriminate the two groups. In multivariable analysis, MSE scale 5 (*p* = 0.010) was the only significantly predictor of severe PH in all HRV variables. In conclusion, the patients with severe PH had worse heart rhythm complexity. MSE parameters, especially scale 5, can help to identify high-risk PH patients.

## 1. Introduction

Pulmonary hypertension (PH) is a progressive, complex, and fatal disease. It involves heterogenous etiologies and different mechanisms [1], and eventually leads to right heart failure. The mortality of PH patients is high even after contemporary treatment [2]; however, timely and intensive management can improve outcomes even in high-risk patients. In addition, the dynamic adjustment of PH medications, based on disease status during follow-up, also plays an important role in PH management [3,4,5]. Therefore, a useful tool for PH risk stratification is urgently needed to guide PH treatment. Several prognostic factors of PH have been verified, including sex, exercise tolerance, right heart hemodynamics, and functional performance [6,7,8], and they have been applied in different prediction models.

In 2015, the European Society of Cardiology (ESC)/European Respiratory Society (ERS) PH guidelines first proposed a dynamic PH risk assessment tool, including a combination of imaging, biologic, hemodynamic, performance status, and clinical conditions [1]. This tool has shown good survival prediction between different risk groups [9,10]; however, it requires right heart hemodynamic measurements, which are invasive and difficult to apply for continuous monitoring of PH severity in clinical practice. Therefore, in this study, we propose a non-invasive and convenient tool for PH risk assessment derived from heart rate variability (HRV), namely, heart rhythm complexity analysis.

Heart rhythm complexity analyzes the complexity of changes in heart rate using non-linear methods, and it has been shown to have better predictive value for the diagnosis of PH and heart failure outcomes [11,12,13] than traditional HRV linear analysis [14]. In our previous study, we found that heart rhythm complexity was decreased in PH patients, and that it was useful to differentiate PH patients from normal populations [13]. However, whether heart rhythm complexity is useful in the risk stratification of PH patients is unknown. Therefore, we designed this study to investigate the clinical application of heart rhythm complexity in the risk stratification of PH patients.

## 2. Materials and Methods

### 2.1. Patients

We prospectively enrolled 54 Taiwanese patients with PH from a single center, including 35 with pulmonary arterial hypertension (PAH) and 19 with chronic thromboembolic pulmonary hypertension (CTEPH) from May 2012 to April 2018. Based on the ESC guidelines [1], the diagnosis of PH was made when the patient had suspicious clinical symptoms, and with mean pulmonary arterial pressure (mPAP) no less than 25 mmHg in right heart catheterization. The World Health Organization (WHO) recognizes five groups of PH, categorized by etiology or comorbidity. The PAH was in the WHO group 1 and CTEPH was in the WHO group 4. The diagnosis of WHO group 1 was made when the pulmonary artery wedge pressure (PAWP) less than 15 mmHg, and pulmonary vascular resistance (PVR) more than 3 Wood Units, and without the evidence of left heart disease. The diagnosis of WHO group 4 was made when the ventilation-perfusion lung scintigraphy showed filling defects in PH patients with the same hemodynamics criteria in right heart catheterization as in PAH. PAH and CTEPH have similar pathophysiological mechanisms as vascular arteriopathy [15], presenting as elevated pre-capillary vessel pressure and pulmonary vascular resistance [16]. Other types of PH may involve complex disease mechanisms, such as lung disease or heart failure, which may result in patient heterogeneity, and were excluded from this study. Therefore, we only enrolled these two PH subgroups in the present study to avoid the confounding influence of other pathophysiologies.

All patients underwent echocardiography, blood sampling, right heart catheterization, and 24-h ambulatory electrocardiogram Holter recording. A full record of medical history of each patient was documented, including dyslipidemia, diabetes mellitus, hypertension, and coronary artery disease. The prescription of PH specific medication was recorded as well. The diagnosis of PH was confirmed by right heart catheterization, based on the ESC guidelines [1]. Parameters, including mPAP, PAWP, right atrial pressure, cardiac output, and cardiac index, were all recorded during right heart catheterization. Blood sampling was obtained during the right heart catheterization. We tested N-terminal pro-brain natriuretic peptide (NT-proBNP), hemoglobin, and creatinine. Both echocardiogram and Holter recordings were performed 2 months before or after right heart catheterization. A six-minute-walk-distance (6MWD) test was recorded 3 months before or after right heart catheterization if the patient was tolerable.

The patients were divided into two groups based on PH severity [17]. The high-risk group was defined as (1) WHO functional class IV or (2) WHO functional class III with severely compromised hemodynamics (right atrial pressure: Pra > 15 mmHg or cardiac index < 2.0 L·min^−1^·m^2^). PH patients in WHO functional class I to II and in WHO functional class III without severely compromised hemodynamics were in the low-risk group [18,19]. There were 20 patients in the high-risk group (group A) and 34 patients in the low-risk group (group B).

This study was approved by the Institutional Review Board of National Taiwan University Hospital (approval numbers NTUH REC No. 201003042R), and all subjects provided written informed consent. All research was performed in accordance with relevant guidelines and regulations. Reporting of this study followed the Strengthening the Reporting of Observational Studies in Epidemiology statement [20].

### 2.2. Echocardiogram

All patients underwent typical transthoracic echocardiography (iE33 x MATRIX Echocardiography System, Philips, Amsterdam, Netherlands). According to the recommendations of the American Society of Echocardiography, tricuspid regurgitation pressure gradient (TRPG) was measured as the peak flow velocity of tricuspid regurgitation (TRV) using a simplified Bernoulli equation: TRPG = 4 × TRV^2^. Left ventricular ejection fraction in M-mode was measured in the parasternal long axis view [21]. The presence of pericardial effusion or not was documented as well.

### 2.3. 24-h Holter Recording and Data Processing

All patients received 24-h ambulatory electrocardiogram Holter recording (Zymed DigiTrak Plus 24-Hour Holter Monitor Recorder and DigiTrak XT Holter Recorder 24-Hour, Philips, Amsterdam, Netherlands) and maintained their original daily activity during the examination without specific limitations. The data were automatically processed using an algorithm and then checked by two technicians. The adopted length of RR Intervals for both linear and non-linear HRV analysis was 4-h and the following criteria was met: (1) between 9 a.m. and 6 p.m.; (2) patients were in awake status; and (3) without sudden increases in heart rate of more than 40 bpm within 1 min to avoid the potential influences of sleep and strong physical activities for both linear and nonlinear analysis, while maintaining enough time length for nonlinear analysis. HRV parameters were processed automatically with MATLAB software.

Nonstationarity can significantly compromise the results of complexity analysis especially for the arrhythmias [22]. We identified the QRS complexes by implementing an adaptive threshold, based on the concept of order-statistic filter, which can be effective for wide ranges and variations of heart rate [23]. Then, the detected QRS peaks were visually inspected to avoid automatic misdetections, and the arrhythmic beats, such as atrial premature contractions, and ventricular premature contractions were removed and replaced by the estimated RR using cubic spline interpolation. Only the RR intervals segments with less than 5% removal were used in this study. In addition, to avoid an unwanted effect of external nonstationarity, we used the empirical mode decomposition method to de-trend the RR series for the oscillation longer than 1 h [24].

### 2.4. Linear HRV Analysis

The interpolated normal-to-normal RR intervals were further used to calculate conventional linear HRV based on the recommendations of the North American Society of Pacing Electrophysiology and the European Society of Cardiology [25]. We analyzed time domain and frequency domain parameters. Time domain analysis included mean RR interval (mean RR), standard deviation of RR interval (SDRR), percentage of absolute differences in normal RR intervals greater than 20 ms (pNN_20_), and percentage of absolute differences in normal RR intervals greater than 50 ms (pNN_50_), representing autonomic nervous system modulation of heart rhythm. The RR intervals were first linearly interpolated at 4 Hz and fast Fourier transform was carried out on the resampled signals. The summation of the power over a different frequency band, including high frequency, (HF, 0.15–0.4 Hz), low frequency (LF, 0.04–0.15 Hz), and very low frequency (VLF, 0.003–0.04 Hz), were calculated as the frequency domain parameters.

### 2.5. Non-Linear HRV Analysis

For non-linear HRV analysis, we applied the multiscale entropy (MSE) and detrended fluctuation analysis (DFA) to quantify the fractal properties of the signals, such as long-term memory effect and information richness over different time-scales. DFA was used to quantify the correlation property of inter-beat interval dynamics in the time series, while eliminating the external nonstationarity by removing the linear-fitted trends in a different time-scale (box-size) [26]. Initially, the average of the normal-to-normal intervals was removed. The resultant signal was then integrated and then divided into segments of equal samples n. The fluctuation F(n) of the signal in the corresponding time-scale n was calculated by the root-mean-square of the integrated signal after removing the fitted trends in the segments. The procedure was then repeated in a different time-scale from a small box-size (e.g., n = 4) to a large box-size (e.g., n = 100). On a double log graph of F(n) and the corresponding box-size (n), the slope of the line was defined as the α exponent, representing the fractal correlation property of the time series. Both short-range (α1: 4–11 beats) and long-range time scales (α2: 11–64 beats) were calculated [27].

MSE analysis is used to measure the complexity of the finite length time series. Compared to a traditional single scale, entropy estimation only measures the degree of regularity on a single time scale; MSE uses “coarse graining” proceeding multiple time scales and provides information richness over different time-scales as the complexity of the system. To estimate entropy, we calculated sample entropy (SampEn) for each coarse-grained time series, and then plotted this as a function of the scale factor. To quantify the complexity of the heartbeat dynamics, in short and long time scales, we calculated the entropy values of scale 5 (scale 5), the linear-fitted slope of scale 1–5 (slope 5), area under the MSE curve for scale 1–5 (area 1–5), and area under the MSE curve for scale 6–20 (area 6–20) [28].

### 2.6. Statistical Analysis

Continuous variables were expressed as mean ± standard deviation for normally distributed variables, and median (interquartile range, 25th and 75th percentiles) for non-normally distributed variables. Categorical variables were expressed as absolute and relative frequencies (percentage). Comparisons were made using the independent *t*-test and the Mann–Whitney U test between two groups. The chi-square test or Fisher’s exact test was used to examine differences between proportions. The discrimination abilities of HRV parameters to high-risk PH were assessed using the receiver operating characteristic (ROC) curve analysis. Logistic regression analysis was used to assess associations between variables and high-risk PH. Significant determinants in univariable logistic regression analysis (*p* < 0.05), including creatinine, PAH group 1, serum creatinine level, plasma NT-proBNP level, mPAP, PVR, LF/HF ratio, DFAα1, slope 1–5, MSE scale 5, and area 6–20, were then tested in multivariable logistic regression analysis with stepwise selection to identify independent factors that could predict high-risk PH. Category-free (continuous) net reclassification improvement (NRI) and integrated discrimination improvement (IDI), were used to evaluate improvements in the accuracy of the prediction after adding a single nonlinear parameter into a model using only linear parameters. NRI is equal to the sum of the increasing probability for survivors and decreasing probability for non-survivors subtracted by the decreasing probability and increasing probability for non-survivors after adopting the updated model. IDI is defined as the average improvement of survival probability for all patients after adopting the updated model [29,30]. The significance of NRI and IDI statistics was based on approximate normal distributions. All statistical analyses were performed using R software 4.0.3 (http://www.r-project.org, accessed on 10 October 2020) and SPSS version 25 for Windows (SPSS Inc., Chicago, IL, USA). The significance level was set at 0.05 (*p* < 0.05).

## 3. Results

### 3.1. Patient Characteristics

The clinical, echocardiographic, and hemodynamic variables of the enrolled patients are listed in Table 1. There were 20 patients in the high-risk group (group A) and 34 patients in the low-risk group (group B).

Compared to group B, there were significantly more patients in group A, in WHO group 1, who had pericardial effusion. In addition, group A had higher levels of serum creatinine and NT-proBNP, and higher TRPG than group B. In pulmonary hemodynamic studies, PVR, and mPAP were significantly higher in group A. The PAH specific medication was listed in Table 1.

### 3.2. Predictors of Interest: HRV Analysis

In linear HRV analysis, group A had significantly lower LF/HF ratio compared to group B. Other linear parameters were comparable between the two groups (Table 2). In non-linear HRV analysis, group A had significantly lower DFAα1, slope 1–5, scale 5 and area 6–20 compared to group B (Table 2). The entropies over different time scales in group A and group B were shown in Figure 1.

#### 3.2.1. Comparisons of Linear and Non-Linear HRV Parameters to Differentiate the High-Risk PH Patients

In ROC curve analysis, MSE scale 5 had highest predictive power to predict the high-risk PH patients. The area under curve (AUC) of MSE scale 5 was 0.758. The AUCs of other linear and non-linear HRV parameters were 0.604 (mean RR), 0.616 (SDRR), 0.560 (pNN_20_), 0.465 (pNN_50_), 0.653 (VLF), 0.579 (LF), 0.460 (HF), 0.682 (LH/HF ratio), 0.681 (DFAα1), 0.510 (DFAα2), 0.671 (slope 1–5), 0.623 (area 1–5), and 0.737 (area 6–20), which were shown in Figure 2.

#### 3.2.2. Logistic Regression Analysis to Predict the Presence of High-Risk PH

In univariable logistic regression analysis, serum creatinine level, PAH, plasma NT-proBNP level, mPAP, PVR, LF/HF ratio, DFAα1, MSE slope 1–5, scale 5, and area 6–20 were significantly associated with the presence of high-risk PH. These parameters were further investigated in multivariable logistic regression analysis, which showed that plasma NT-proBNP levels (odds ratio [OR]: 1.001, 95% confidence interval [CI]: 1.000~1.002, *p* = 0.009), and MSE scale 5 (OR: 0.009, 95% CI: <0.001~0.324, *p* = 0.010) were remained in the model and both NT-proBNP level and MSE scale 5 were significantly associated with the presence of high-risk PH (Table 3).

#### 3.2.3. The Effect of Adding Heart Rhythm Complexity to the Linear HRV Parameters to Identify High-Risk PH Patients

In both NRI and IDI models, the MSE scale 5 significantly improved the discrimination power of all linear HRV parameters, including mean RR, SDRR, VLF, LF, HF, and LF/HF ratio. Area 6–20 significantly improved the discrimination power of mean RR, VLF, and HF in both NRI and IDI models, and SDRR, LF, and LF/HF ratio in IDI model. DFAα1 significantly improved the discrimination power of SDRR, VLF, LF, and HF in both the NRI and IDI models, and mean RR in the IDI model (Table 4).

## 4. Discussion

The main finding of this study was that heart rhythm complexity was significantly depressed in high-risk PH patients. In addition, adding heart rhythm complexity predictors to traditional linear HRV parameters improved the power to predict high-risk PH patients. This is the first study to demonstrate an association between heart rhythm complexity and severity of PH, and the better performance of heart rhythm complexity in identifying high-risk PH patients than traditional HRV parameters.

PH is a critical disease, which needs an early diagnosis and timely management. Patients classified as being at high risk according to the 2015 ESC/ERS PH guidelines have a worse prognosis compared to patients at low risk. Sitbon et al. demonstrated that poor functional status was associated with poor outcomes. In their study, PH patients in WHO functional class IV and those in class III with severely compromised hemodynamics had the worst outcomes [19]. Previous studies have demonstrated that early interventions including both pharmacological and multidisciplinary team care can improve the outcomes of PH patients, even those with severe disease and poor functional status [5]. Therefore, identifying high-risk patients is essential for the management of PH. Several survival prediction models have been proposed for PH patients; however, they are complex and difficult to use [31]. Currently, the 2015 ESC/ERS PH guidelines advocate assessing the risk of PH by using a combination of several different tools, and this method is widely used in daily practice [1]. However, risk assessment requires invasive right heart catheterization, which is difficult to apply in frequent monitoring during follow-up. Therefore, there is still a strong unmet need for an easy-to-use tool to allow for both timely and continuous monitoring of disease status to improve the clinical care of PH patients.

HRV is a useful non-invasive tool, which has been studied in many diseases, including coronary artery disease, heart failure, and even pulmonary hypertension [32,33,34]. It has been correlated with autonomic dysfunction and used as an outcome predictor. Porte et al. demonstrated that heart rate complexity parameters decreased due to sympathetic activation during postural change [35]. Another study showed that sympathetic activation during senescence was associated with impaired heart rate complexity [36]. These evidences supported that the usefulness of heart rate complexity in monitoring sympathovagal balance. Pulmonary hypertension is characterized by increased pulmonary artery pressure leading to right ventricular failure [37]. The serum norepinephrine increased in patients with PAH similar to those with congestive heart failure as the indicator of cardiac sympathetic activation [38]. Furthermore, sympathetic activation has also been correlated with the severity of PH [39,40]. Several studies also showing that measuring autonomic system regulation using HRV could be a predictor of disease severity and long-term outcomes in PH [41,42,43,44]. Since that, overactivation of sympathetic systems is likely to be one of the major reasons explaining the worse HRV and complexity in severe PH patients. Bienias et al. demonstrated that patients with arterial or chronic thromboembolic PH had significantly impaired heart rate turbulence, a linear HRV parameter [45]. Recently, Peng et al. proposed the heart rhythm complexity derived from two non-linear parameters of HRV, DFA, and MSE, to evaluate complexity change in the biological systems [26,28,46]. Heart rhythm complexity was shown to have better efficacy and predictive power for various diseases than traditional HRV [14,47].

Heart rhythm complexity measures the complexity of changes in the R–R interval, which contains detailed information derived from heart rate dynamics. Once a biological system has become diseased, the complexity breaks down, and non-linear HRV analysis, can detect subtle changes at an early stage [48]. In a retrospective study, abnormal DFAα1 in asymptomatic heart failure patients was associated with the onset of heart failure years in advance of the first clinical event [49,50]. Tsai et al. recently demonstrated that heart rhythm complexity had a better prognostic value for cardiovascular events in patients undergoing peritoneal dialysis compared with linear HRV analysis [47]. In recent years, heart rhythm complexity was extensively studied in many fields, including left heart failure [51], post-infarction myocardial function [52], patients undergoing dialysis [12,47,53], severity of abdominal aorta calcification [54], primary aldosteronism [24], stroke [55], and PH [56]. These studies support the importance of heart rhythm complexity in clinical practice and its potential role in disease risk stratification. In the present study, we demonstrated that heart rhythm complexity parameters, especially MSE scale 5, were significantly associated with PH disease severity and could be used in PH risk stratification. To the best of our knowledge, this is the first study to apply heart rhythm complexity to the prediction of PH disease severity. Although the improvement of the complexity can be attribute to not only the enhanced complexity characteristics but the magnitude of HRV [57], combining different parameters of MSE can give us better information related to the “quality” (complexity) or the “quantity” (magnitude of HRV) changes [58]. Furthermore, model-free complexity can assess the embedded space with variable scales grouped by the K-nearest-neighbor to avoid coarse-graining that may introduce bias due to the fixed dimensions as well as the aliasing filter effect [57,59]. Recently, the local version of the sample entropy was proposed to eliminate the nonstationary effect on the results of complexity analysis [60]. The research by using those new methods warrants further study. In addition, the cardiopulmonary coupling is another important issue in the HRV analysis focusing on the interaction between cardiovascular and cardiorespiratory systems. The cardiorespiratory coupling between the systems is thought to be with each other in a nonlinear way [61]. MSE has been used to evaluate the cardiorespiratory coupling and the asynchrony. Platiša et al. demonstrated that primary alterations in the regularity of cardiac rhythm resulted in changes in the regularity of the respiratory rhythm in heart failure patients [62]. However, there were limited studies investigating the cardiorespiratory coupling in PH patients. Further studies may be needed to integrate the role of cardiorespiratory coupling in PH patients.

Compared with heart rhythm complexity, linear HRV parameters, including SDRR, SDRR index, VLF, LF/HF ratio, and heart rate turbulence have been widely studied to assess PH [45,63]. Recent studies have also demonstrated an association between impaired linear HRV parameter, SDRR, and PH disease severity markers, including impaired WHO functional status, decreased 6MWD, impaired tricuspid annular plane systolic excursion, right ventricular systolic function, higher TRPG, and NT-proBNP level [64,65,66]. In this study, we first demonstrated a better association between heart rhythm complexity and PH disease severity compared to traditional HRV analysis. Second, the discrimination power of linear HRV for PH disease severity improved significantly after combining heart rhythm complexity parameters. The combination of linear and non-linear HRV parameters to form a new predictive model may have further improved its risk stratification ability and outcome prediction.

There are several limitations to this study. First, this is a pilot study. The number of cases was small, and further studies are needed to validate the results. In addition, model-free complexity analysis or entropy with local characteristics can preserve more information instead of a one-fit-all algorithm. Those methods should be included for a large-scale study to probe the underlying pathophysiological mechanisms related to the changes of the complexity of the PH patients. Second, we only enrolled PH patients in WHO group 1 and group 4, and future studies should enroll different groups of PH patients to investigate the potential application of HRV in these patients. Third, this pilot study is a cross-sectional design and lacks clinical long-term follow-up data. A prospective cohort study with clinical end-point follow-up is needed to confirm the utility of heart rhythm complexity on clinical outcome predictions.

## 5. Conclusions

This study demonstrated that high-risk PH patients had worse heart rhythm complexity. MSE scale 5 had the best discrimination power to predict high-risk PH patients. Moreover, adding MSE scale 5, area 6–20 or DFAα1 to linear HRV parameters significantly improved the predictive power for high-risk PH patients. Heart rhythm complexity can potentially be used as (i) an indicator of PH disease severity, and (ii) to stratify the risk of PH.

## Figures and Tables

**Figure 1 entropy-23-00753-f001:**
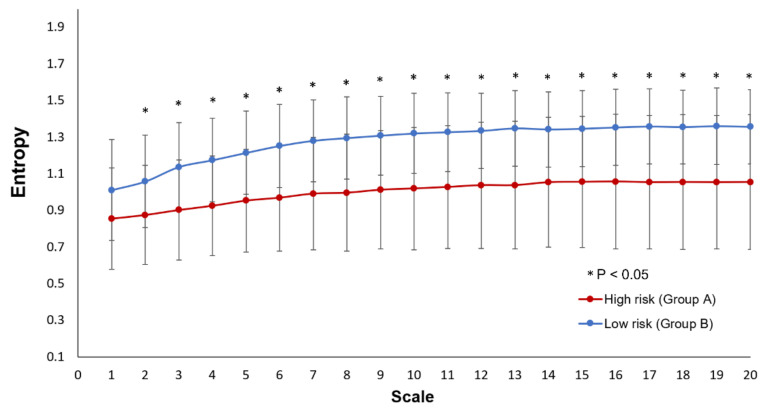
The entropy over different time scales in patients in high-risk group (Group A: red) and low-risk group (Group B: blue). * *p* < 0.05, comparing entropy at different scale between high-risk and low-risk PH patients with independent T test.

**Figure 2 entropy-23-00753-f002:**
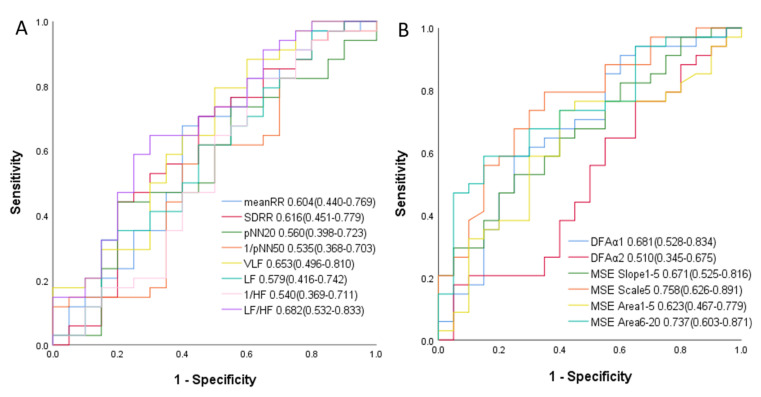
Analysis of the discrimination power of HRV variables for PH risk stratification using receiver operating characteristic curve analysis. (**A**) ROC curves by using linear HRV parameters for predicting high-risk PH patients; (**B**) ROC curves by using heart rhythm complexity parameters for predicting high-risk PH patients. (Abbreviations: Mean RR, mean RR interval; SDRR, standard deviation of RR interval; pNN_20_, percentage of absolute differences in normal RR intervals greater than 20 ms; pNN_50_, percentage of absolute differences in normal RR intervals greater than 50 ms; VLF, very low frequency; LF, low frequency; HF, high frequency; DFA, detrended fluctuation analysis; MSE, multiscale entropy.

**Table 1 entropy-23-00753-t001:** Clinical Data of the patients.

	High-Risk Group (N = 20)	Low-Risk Group (N = 34)	*p* Value
Age (years)	43.80 ± 10.70	45.76 ± 11.34	0.533
Male, n (%)	9 (45%)	12 (35%)	0.480
BMI (kg·m^−2^)	22.09 ± 3.85	24.21 ± 4.41	0.081
CAD, n (%)	1 (5%)	1 (3%)	1.000
DM, n (%)	2 (10%)	3 (9%)	1.000
HTN, n (%)	1 (5%)	5 (15%)	0.395
Dyslipidemia, n (%)	1 (5%)	3 (9%)	1.000
PAH (WHO group 1)	17 (85%)	18 (53%)	0.017
Hemoglobin (g/dL)	13.72 ± 3.15	13.52 ± 3.76	0.835
Creatinine (mg/dL)	1.15 ± 0.67	0.76 ± 0.26	0.024
Log NT-proBNP	3.34 ± 0.54	2.52 ± 0.54	<0.001
NT-proBNP (ng/dL)	1510 (959~6428)	292 (116~1045)	<0.001
LVEF (%)	68.55 ± 9.46	68.62 ± 10.07	0.977
TRPG (mmHg)	93.31 ± 31.8	64.67 ± 28.10	0.001
Pericardial effusion, n (%)	7 (35%)	1 (3%)	0.003
6MWD (m)	298.31 ± 128.00	367.42 ± 120.32	0.074
mPAP (mmHg)	58.11 ± 15.46	47.44 ± 15.27	0.021
PVR (Wood Units)	13.63 ± 6.00	8.24 ± 4.23	0.002
CO (L·min^−1^)	3.71 ± 1.59	4.45 ± 1.30	0.081
CI (L·min^−1^·m^2^)	2.26 ± 0.97	2.75 ± 0.86	0.069
PAWP (mmHg)	14.00 ± 4.23	12.09 ± 3.69	0.097
PAH specific medication			
Sildenafil, n (%)	8 (40%)	15 (44%)	0.768
Macitentan, n (%)	3 (15%)	1 (3%)	0.138
Riociguat, n (%)	0 (0%)	6 (18%)	0.074
Bosentan, n (%)	2 (10%)	2 (6%)	0.622
Iloprost, n (%)	4 (20%)	1 (3%)	0.057
Epoprostenol, n (%)	1 (5%)	1 (3%)	1.000

Abbreviation: BMI, body mass index; CAD, coronary artery disease; DM, diabetes mellitus; HTN, hypertension; PAH, pulmonary arterial hypertension; NT-proBNP, N-terminal Pro-Brain Natriuretic Peptide; LVEF, left ventricular ejection fraction; TRPG, tricuspid regurgitation pressure gradient; 6MWD, 6-min-walk-distance; mPAP, mean pulmonary artery pressure; PVR, pulmonary vascular resistance; CO, cardiac output; CI, cardiac index; PAWP, pulmonary arterial wedge pressure.

**Table 2 entropy-23-00753-t002:** Holter parameters of the patients.

	High-Risk Group (N = 20)	Low-Risk Group (N = 34)	*p* Value
Time Domain Analysis			
Mean RR (ms)	684.03 (605.77~795.63)	748.63 (678.30~805.53)	0.203
SDRR (ms)	57.14 (43.84~65.88)	64.42 (54.37~87.43)	0.162
pNN_20_ (%)	19.17 (9.20~26.67)	20.86 (13.94~36.88)	0.463
pNN_50_ (%)	3.47 (0.32~12.32)	2.21 (0.77~6.64)	0.667
Frequency Domain Analysis			
VLF (ms^−2^)	172.56 (46.43~543.01)	384.16 (169.56~604.98)	0.062
LF (ms^−2^)	64.99 (19.52~140.02)	98.00 (38.11~174.58)	0.333
HF (ms^−2^)	42.28 (12.81~227.52)	36.46 (15.94~125.03)	0.629
LF/HF ratio	1.06 (0.56~2.17)	2.14 (1.03~3.61)	0.026
Detrended fluctuation analysis			
DFAα1	0.92 (0.56~1.05)	1.04 (0.89~1.23)	0.028
DFAα2	1.12 (1.01~1.19)	1.11 (1.03~1.17)	0.900
Multiscale entropy			
Slope 1–5	−0.008 (−0.075~0.039)	0.04 (−0.03~0.07)	0.038
Scale 5	1.01 (0.73~1.14)	1.22 (1.06~1.36)	0.002
Area 1–5	3.30 (2.94~4.44)	4.18 (3.26~4.89)	0.135
Area 6–20	15.94 (12.48~18.40)	18.89 (15.16~20.91)	0.004

Abbreviation: Mean RR, mean RR interval; SDRR, standard deviation of RR interval; pNN_20_, percentage of absolute differences in normal RR intervals greater than 20 ms; pNN_50_, percentage of absolute differences in normal RR intervals greater than 50 ms; VLF, very low frequency; LF, low frequency; HF, high frequency; DFA, detrended fluctuation analysis; area 1–5, area under the MSE curve for scale 1–5; area 6–20, area under the MSE curve for scale 6–20.

**Table 3 entropy-23-00753-t003:** Univariable and multivariable logistic regression model to predict the high-risk group in pulmonary hypertension.

Univariable Logistic Regression			Multivariable Logistic Regression	
	OR (95% CI)	*p* Value	OR (95% CI)	*p* Value
Age (Year)	0.984 (0.935~1.035)	0.525		
Sex (man)	1.500 (0.486~4.631)	0.481		
BMI (kg·m^−2^)	0.884 (0.768~1.017)	0.086		
PAH group 1	5.037 (1.242~20.43)	0.024		
Creatinine (mg/dL)	8.301 (1.358~50.75)	0.022		
NT-ProBNP (ng/dl)	1.001 (1.000~1.002)	0.019	1.001 (1.000~1.002)	0.009
6MWD (m)	0.995 (0.990~1.001)	0.080		
mPAP (mmHg)	1.046 (1.005~1.089)	0.029		
CI (L·min^−1^·m^2^)	0.525 (0.258~1.067)	0.075		
PVR (Wood Units)	1.232 (1.070~1.418)	0.004		
Mean RR (ms)	0.997 (0.992~1.002)	0.198		
SDRR (ms)	0.992 (0.973~1.010)	0.373		
pNN20 (%)	0.993 (0.961~1.025)	0.647		
pNN50 (%)	1.016 (0.971~1.063)	0.503		
VLF (ms^−2^)	0.998 (0.996~1.000)	0.081		
LF (ms^−2^)	0.999 (0.997~1.002)	0.543		
HF (ms^−2^)	1.000 (0.999~1.001)	0.858		
LF/HF ratio	0.622 (0.391~0.990)	0.045		
DFAα1	0.072 (0.008~0.626)	0.017		
DFAα2	0.457 (0.006~33.761)	0.721		
Slope 1–5	0.000 (0.000~0.560)	0.036		
Scale 5	0.012 (0.001~0.222)	0.003	0.009 (<0.001~0.324)	0.010
Area 1–5	0.705 (0.418~1.189)	0.190		
Area 6–20	0.835 (0.714~0.977)	0.024		

Abbreviation: BMI, body mass index; CAD, coronary artery disease; DM, diabetes mellitus; HTN, hypertension; PAH, pulmonary arterial hypertension; NT-proBNP, N-terminal Pro-Brain Natriuretic Peptide; LVEF, left ventricular ejection fraction; TRPG, tricuspid regurgitation pressure gradient; 6MWD, 6-min-walk-distance; mPAP, mean pulmonary artery pressure; PVR, pulmonary vascular resistance; CO, cardiac output; CI, cardiac index; PCWP, pulmonary capillary wedge pressure; Mean R-R, mean RR interval; SDRR, standard deviation of RR interval; pNN_20_, percentage of absolute differences in normal RR intervals greater than 20 ms; pNN_50_, percentage of absolute differences in normal RR intervals greater than 50 ms; VLF, very low frequency; LF, low frequency; HF, high frequency; DFA, detrended fluctuation analysis; area 1–5, area under the MSE curve for scale 1–5; area 6–20, area under the MSE curve for scale 6–20.

**Table 4 entropy-23-00753-t004:** AUC, NRI, and IDI models of linear parameters before and after adding DFAα1 and MSE parameters for risk stratification in pulmonary hypertension.

Parameters	AUC	R Square	NRI	NRI *p* Value	IDI	IDI *p* Value
Mean RR	0.604	0.032				
+Scale5	0.775	0.051	0.694	0.008	0.194	0.001
+Area 6–20	0.749	0.12	0.535	0.048	0.092	0.026
+DFAα1	0.701	0.126	0.494	0.071	0.095	0.028
SDRR	0.615	0.015				
+Scale5	0.781	0.12	0.771	0.003	0.211	0.001
+Area 6–20	0.731	0.121	0.494	0.071	0.107	0.014
+DFAα1	0.681	0.123	0.535	0.048	0.108	0.017
VLF	0.653	0.061				
+Scale5	0.782	0.117	0.535	0.048	0.171	0.002
+Area 6–20	0.725	0.147	0.653	0.014	0.082	0.035
+DFAα1	0.699	0.145	0.694	0.008	0.084	0.037
LF	0.579	0.008				
+Scale5	0.768	0.086	0.771	0.003	0.209	0.001
+Area 6–20	0.731	0.118	0.494	0.071	0.112	0.012
+DFAα1	0.694	0.134	0.553	0.042	0.129	0.01
HF	0.54	0.001				
+Scale5	0.76	0.029	0.871	0.001	0.221	<0.001
+Area 6–20	0.734	0.116	0.553	0.042	0.118	0.01
+DFAα1	0.694	0.129	0.612	0.023	0.132	0.009
LF/HF ratio	0.682	0.075				
+Scale5	0.806	0.077	0.771	0.003	0.184	0.001
+Area 6–20	0.76	0.156	0.394	0.154	0.068	0.039
+DFAα1	0.718	0.114	0.335	0.228	0.027	0.19

Abbreviation: Mean RR, mean RR interval; SDRR, standard deviation of RR interval; VLF, very low frequency; LF, low frequency; HF, high frequency; DFA, detrended fluctuation analysis.

## Data Availability

The data presented in this study are available on request from the corresponding author. The data are not publicly available due to privacy.

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
