# Peer review of "The Value of Heart Rhythm Complexity in Identifying High-Risk Pulmonary Hypertension Patients"

_entropy, 2021, doi:10.3390/e23060753_

Round 1

Reviewer 1 Report

The paper from Tang et. al. is well-grounded and addresses the importance of heart rate complexity in the risk stratification of diseases. However, the text is too concise and important information is missing. A thorough revision of the text should be conducted, especially in “Material and Methods”.

General Comments

  • The paper is lacking a discussion on the physiological effects of PH on the autonomic nervous system.
  • Is PH expected to change respiration? Several studies reported the importance of MSE scale 5 (including Costa’s original paper), which may be linked to changes over respiratory cycles. The authors could discuss that.

Specific comments

  • Section 2: the title of the section could be only “Material and Methods”.
  • Line 68: PAH was not defined. Also, what are “WHO groups 1” and “WHO group 4”? Please, describe them.
  • Line 81: what are the “WHO functional classes”? Are they different from NYHA functional classes?
  • The division criteria of patients in groups A and B are not clear. For example, if the patient is in the functional class II or IV but does not have Pra > 15 nor CI < 2, is it classified in group A or B? Also, if the patient is in functional class I or II but has one of the above hemodynamic alterations, which class is it classified into?
  • Lines 91-93: TRGP and TRV were not defined.
  • Lines 108-109: “including the calculation of HF, LF and VLF powers…”. Also, please be more specific about the methodology adopted for frequency domain analysis. For example, did you interpolate the series prior to FT? Did you use the Welch protocol? If yes, describe all the parameters adopted.
  • Lines 112-113: why are MSE and DFA related to fractal and stochastic theories, respectively?
  • Lines 117-123: this definition of DFA lacks mentioning that the average is removed before integration. Moreover, the trending within each box is removed before RMS calculation and the relationship between RMS and box size is what represents the fractal scaling.
  • Lines 125-126: “Traditional single scale entropy estimation yields lower entropy in times series.“ I did not understand this sentence.
  • Line 130: “… calculated entropy at scale 5…”?
  • Line 140: several variables mentioned here were not previously introduced in “Methods”. How were they collected? In which timeframe from the other exams? Etc.
  • Line 143: what are NRI and IDI? Please, describe those methods and provide references.
  • Table I: most of the variables described here were not described in “Methods”. Please, introduce them in “Methods”.
  • Figure 1: please, include the error bars for each scale.
  • Lines 187-188: a reference to a figure in the subsection title is not usual.
  • Figure 2: in the caption, you describe some acronyms that were not evaluated (TRGP, PVR, NT-proBNP, 6MWD).
  • Section 3.2.2: what is the predictive power for the combination of NT-pro BNP and MSE scale 5? This is not clear.
  • Table 4: since the outcome is the risk class (categorical variable), how is R squared estimated? Moreover, it is surprising that AUC is high in most of the situations (>0.7) while R squared is too low at the same time (<0.15).
  • Line 280: “… DFA and MSE, which are based on fractal and chaos therapy”. Please, revise this sentence. Why do you consider MSE is based on chaos theory?
  • Section 6 (“Patents”) is empty.

Author Response

Dear Reviewer,
 We appreciate your excellent opinions and advice. The comments of the reviewers have been addressed point by point in this letter. Modifications have been typed in blue color in the revised version. Please see the attachment for the revised manuscript and response letter.
Thank you for your kind reprocessing of our manuscript.
Sincerely Yours

Yen-Hung Lin, M.D., Ph.D.
Department of Internal Medicine
National Taiwan University Hospital

Reviewer 2 Report

The study assesses the influence of ectopic beats on heart rate variability (HRV) analysis in time, frequency and information domains.

The study is interesting and well written. However, some choice made by the authors should be better justified and some issues need to be deeply discussed.  

  • The selection of the HRV complexity metrics utilized in this study might appears to be arbitrary. Alternative list of HRV complexity metrics can be found in A. Porta et al, IEEE Trans Biomed Eng, 64, 1287-1296, 2017 and A. Porta et al, IEEE Trans Biomed Eng, 66, 623-631, 2019). Please justify the choice and discuss alternative possibilities that might improve further the proposed classification.
  • The selection seems to be arbitrary given that one of the exploited tools, namely multiscale entropy (MSE) is a suboptimal approach because the filtering procedure applied to select time scales does not assure relevant anti-aliasing properties and the estimator tends to mix effects of complexity with those of the magnitude of HRV (see JF Valencia et al, IEEE Trans Biomed Eng, 56, 2202-2213, 2009). Please discuss the impact of this bias over the conclusions of this study or apply the less bias estimator given in the abovementioned paper.
  • It is well-know that HRV complexity decreased during an acute sympathetic activation (see A. Porte et al, J Appl Physiol, 103, 1143-1149, 2007) or during a chronic increase of the sympathetic drive such as during senescence (see A.M. Catai et al. Entropy, 16, 6686-6704, 2014). It seems to me that a strong sympathetic drive reducing HRV complexity could be the main drive of these results and the ability of complexity markers in stratifying the risk in pulmonary hypertensive patients. Discussion could be integrated at this regard.
  • The study tends to mix methods assessing short-term cardiac neural control with those assessing long-term HRV regulation. Please report explicitly the length of the series for each analysis and justify the adopted length.
  • Nonstationarities, such as those induced by artifacts and arrhythmias, are known to affect frequency and complexity markers derived from HRV (see V. Magagnin et al, Physiol Meas, 32, 1775-1786, 2011). It is unclear how the authors addressed this issue. Was stationarity tested? Were HRV series preprocessed to limit the impact of nonstationarities? How were isolated ectopic beats treated? Could nonstationarities affect conclusions of this study above and beyond the value of complexity markers? The authors should discuss deeply these issues.

Author Response

(The authors gave the same response as above.)

Round 2

Reviewer 1 Report

The authors properly addressed all my comments. I have only two very minor writing points:

Line 370: symptomatic à sympathetic.

Line 371: associated with.

Reviewer 2 Report

The manuscript was improved. The authors replied satisfactorily to all my issues and followed carefully the suggestions given.

This manuscript is a resubmission of an earlier submission. The following is a list of the peer review reports and author responses from that submission.